# A Multiparameter Gas-Monitoring System Combining Functionalized and Non-Functionalized Microcantilevers

**DOI:** 10.3390/mi11030283

**Published:** 2020-03-10

**Authors:** Christof Huber, Maria Pilar Pina, Juan José Morales, Alexandre Mehdaoui

**Affiliations:** 1TrueDyne Sensors AG, 4153 Reinach BL, Switzerland; alexandre.mehdaoui@truedyne.com; 2Nanoscience Institute of Aragon (INA), University of Zaragoza, 50009 Zaragoza, Spain; jjmorale@unizar.es; 3Instituto de Ciencia de Materiales de Aragon (ICMA), Universidad de Zaragoza-CSIC, 50009 Zaragoza, Spain

**Keywords:** microcantilever, nanoporous functional coatings, welding gas monitoring, ppm of water content

## Abstract

The aim of the study is to develop a compact, robust and maintenance free gas concentration and humidity monitoring system for industrial use in the field of inert process gases. Our multiparameter gas-monitoring system prototype allows the simultaneous measurement of the fluid physical properties (density, viscosity) and water vapor content (at ppm level) under varying process conditions. This approach is enabled by the combination of functionalized and non-functionalized resonating microcantilevers in a single sensing platform. Density and viscosity measuring performance is evaluated over a wide range of gases, temperatures and pressures with non-functionalized microcantilevers. For the humidity measurement, microporous Y-type zeolite and mesoporous silica MCM48 are evaluated as sensing materials. An easily scalable functionalization method to high-throughput production is herein adopted. Experimental results with functionalized microcantilevers exposed to water vapor (at ppm level) indicate that frequency changes cannot be attributed to a mass effect alone, but also stiffness effects dependent on adsorption of water and working temperature must be considered. To support this hypothesis, the mechanical response of such microcantilevers has been modelled considering both effects and the simulated results validated by comparison against experimental data.

## 1. Introduction

The aim of the presented sensor project is to develop a compact, robust and maintenance-free gas-monitoring system for industrial use in the field of inert gases e.g., welding gas or modified atmospheric packaging gas mixing application. In such applications typically binary or ternary mixtures of argon, helium, nitrogen, carbon dioxide, oxygen or hydrogen are used. Gas concentrations must be controlled with accuracies in the range of 1%. In most cases, the humidity must also be monitored in these applications. As an example, shielding gas supplies are controlled to very low moisture content (−57 °C dewpoint or lower). Typical threshold values are between 200 and 40 ppm for welding gases [1]. Moisture (H_2_O) is a prime source of hydrogen. At arc temperatures, water breaks down releasing hydrogen atoms that cause porosity in weldments. 

The state of the art of technology is to connect several independent sensors in series, e.g., thermal conductivity, together with specific optical absorption and dew point sensors. Such installations are bulky and require frequent recalibration. Advances in the field of micro-electro-mechanical systems (MEMS) now offer unique opportunities to design sensitive and cost-effective analytical platforms. To bring the same functionality in one multiparameter sensor system, we combine functionalized and non-functionalized microcantilevers working in dynamic mode with electromagnetic actuation and piezoresistive detection, together with a pressure and a temperature sensor on one printed circuit board (PCB) that is exposed to the process gas. The project was presented the first time at the 4^th^ conference on microfluidic handling systems 2019 [2], and here we present an extended, more elaborate overview of our research. Special efforts were focused on the development of a cost-effective functionalization process for large-scale manufacturing. By using these MEMS as transducers, high sensitivity is guaranteed; and the problem of achieving selective sensing (i.e., discrimination of the target compound among other components in a mixture) is overcome by means of a proper functionalization of the cantilever surface or by switching the operation mode (static vs. dynamic) and working temperature. Any fluctuations of the microcantilever temperature that, in turn, may lead to parasitic piezoresistance changes and frequency shifts is eliminated by using the bare microcantilever as reference sensor. 

In this study, we focus on the feasibility of humidity measurement at ppm level on synthetic welding gas mixtures with functionalized micro-cantilevers in an industrial environment, which means that gas composition, temperature and pressure may vary and, therefore, the sensor must be able to work under non-ideal conditions. This is in contrast to most previous studies on this topic, where the behaviour of the sensors was investigated under very stable and controlled conditions [3,4,5,6]. 

The goals of this paper are to stablish the optimal conditions for the functionalization of microcantilevers with hydrophilic materials submicrometric in size, to characterize their humidity sensing performance at ppm level and to explore their applicability, combined with bare microcantilevers, as a multiparameter gas-monitoring system for synthetic welding gas mixtures. Furthermore, a mathematical model of the mechanical response of the resonating microcantilevers when exposed to ppm of water has been developed to provide useful guidelines for the definition of the monitoring routines.

## 2. Materials and Methods 

### 2.1. Experimental Setup

#### 2.1.1. Sensor Printed Circuit Board (PCB) and Gas-Measuring Chamber

Commercially available silicon microcantilevers initially designated for atomic force microscopy from SCL-Sensor.Tech. Fabrication GmbH (1220 Vienna, Austria) [7] are used (see Figure 1). The cantilevers have a length of 300 µm a width of 110 µm and a thickness between 2.5 to 4 µm. The top surface of Si beam consists of three different layers: a 200 nm-thick SiO_2_ layer, the Al heater tracks of 600–800 nm thickness and an Al_2_O_3_ insulation layer on top. The hydrophilicity and roughness of the final Al_2_O_3_, in the range of 100 nm could potentially contribute to water sorption under certain conditions. The use of porous aluminium anodic oxide on cantilevers is successfully demonstrated for measurement of moisture in most industrial gases [5]. The resonance frequencies of the microcantilevers are in the range of 30 to 60 kHz and quality factors of 50 to 200. The cantilevers have a heater Al coil on the tip that can be used to heat the cantilever as well as to drive the cantilever by Lorenz force actuation. The detection of the cantilever motion is undertaken by piezoresistive sensing probes. The cantilevers are mounted on a small cantilever PCB with a 10-pin connector on its lower side. In our setup, the bare cantilever is used to measure the density and the viscosity of the gas as reported by Badarlis et al. [8] or Huber et al. [9]; and the functionalized counterpart is used to measure water vapor at ppm level thanks to the hydrophilic porous layer deployed on top of the cantilever surface, as reported for example by Urbiztondo et al [10]. 

For the measurement in gases, we have manufactured a special PCB which is equipped on both sides as can be seen in Figure 2. It contains on each side a 10-pin connector which can be used to connect the cantilever PCBs. In front of the cantilevers a permanent magnet is mounted which serves the Lorenz force excitation of the cantilever. Additionally, the PCB contains a pressure and temperature sensor with I2C output [11]. To prevent self-heating by electronic components, the PCB does not contain any active electronic elements, but all traces are led out via a 4-pin and two 8-pin M8 connectors. The PCB is screwed into a gas-tight metal cylinder with radial fluidic port connections that constitutes the measuring chamber with a total free volume circa 20 cm^3^ (see Figure 2). The gas measuring chamber allows measurements up to pressures of at least 10 bar.

#### 2.1.2. Signal Processing 

The signal processing equipment is placed outside the gas measuring chamber (Figure 3). The cantilevers are driven and measured with the help of a Universal Resonance Analyzer (MFA200) from MicroResonant OG (Linz, Austria) [12]. The MFA200 provides an excitation signal generator, a response signal analyser and a digital signal-processing stage extracting the parameters of the resonator from excitation and response signals. The evaluation unit and algorithms used have been described in detail in several publications [13,14] and are no longer specifically discussed here. The electronic readout interface provides the resonance frequency and the quality factor of the 2 cantilevers as well as the pressure and the temperature in the measuring chamber. The MFA 200 firmware was adapted so that the two cantilevers can be alternately measured. In addition, a heater function is included, which means that a direct current (DC) voltage pulse can be superimposed on the alternating current (AC) voltage excitation signal during operation. This DC voltage pulse provokes the heating of the cantilever without interrupting the measurement and enables the sensing layer conditioning. In our previous work with zeolite-coated microcantilevers [10], the influence of degassing conditions on the sensing performance is thoroughly discussed. An adequate DC voltage has to be supplied to the heating resistor to increase the temperature of the support (above 100 °C) directly under the sensing coating before and after any gas-sensing measurement in order to release the nanopores. The supplied heating power has to be defined in accordance with the sorbent–sorbate interactions (see Section B.1.). In this work, the microcantilevers were typically degassed at temperatures > 200 °C by consecutive heating cycles, circa 20 s duration each. It is worth mentioning that the temperature values provided in this work have been calculated from those measured on the PCB. Some assumptions are needed to correlate it with the sensing layer and the surrounding gas atmosphere with specific thermal conductivity and heat capacity values. As an example, a heating power of 50 mW leads to local temperatures on the cantilever tip varying from 200 to 250 °C when exposed to air under atmospheric pressure. In addition, the exact power supplied to the chip is not measured, which in turn, hinders accurate temperature evaluation of the chip. Additional efforts on the heating unit are still required for a practical implementation.

#### 2.1.3. Fluidic Measurements Setup

Figure 3 shows a schematic representation of the whole measurement setup used for the experiments. The measuring chamber could be fed with 5 different gases (Air, Ar, N_2_, CO_2_ and He). The measurements with humidity content were carried out with a certified gas mixture 100 ppmV H_2_O content in Ar supplied by Nippon Gases Europe. The total pressure in the measuring chamber could be varied in the range from 1 to 10 bar using a pressure regulation valve. The temperature of the measuring chamber was controlled by a jacket flushed with water from a Julabo thermostatic bath. In the measuring chamber, a dew point sensor from CS Instrument (FA510, CS Instruments GmbH & Co. KG, Harrislee, Germany) [15] was installed for measuring the humidity.

### 2.2. Cantilever Functionalization 

#### 2.2.1. Hydrophilic Nanoporous Materials

Commercial microporous Y type zeolite CBV100 supplied by Zeolyst (Zeolyst International, Conshohocken, PA, US) (Si/Al ratio of 2.55, Na as the extra-framework cation,), and mesoporous silica MCM-48, synthesized according to our previous work [16], were investigated for this study. The textural characterization was determined from Ar/N_2_ adsorption isotherms measured on a Micromeritics ASAP 2020. The samples were previously degassed under vacuum. BET analysis was performed using the appropriate pressure range based on published consistency criteria and pore size and pore volume were determined by using Horvath–Kawazoe analysis of the isotherm [17]. Figure 4, Figure 5 and Table 1 show the morphological and textural characterization of the hydrophilic materials selected for this work.

The water sorption properties were estimated from thermogravimetric analyses on a thermo-balance PERSEUS® STA 449 F3 Jupiter® from Netzsch combined with a Modular Humidity Generator ProUmid MHG32 capable for relative humidity control (herein varied from 0.5% to 3.5% relative humidity (RH) at 44 °C) in the sample chamber. Firstly, the samples were in situ degassed with N_2_ (99.999% purity) as sweep gas (150 mL/min) at 200 °C using a heating rate of 10 °C/min for 3 h. After cooling at 40 °C or 60 °C, and signal stabilization (considered as initial weight of the sample) water sorption experiments were launched (see Figure 6, [18] and also Appendix B). It can be seen, that the dependency of the adsorbed amount with partial pressure or water concentration (vapor ratio) is linear below 2500 ppmV. 

#### 2.2.2. Functionalization Method 1: Oxygen Plasma and Incubation 

Two different methods were developed and tested to apply the inorganic particles, submicrometric in size, to the cantilever surface. In the first, prior to the chip incubation step in the aqueous suspension 2% wt of particles, the top surface of the microcantilever was activated by O_2_ plasma to promote the self-assembly of the particles onto the surface. The optimal experimental conditions for the sensing materials herein investigated have been defined from the experimentation with Si/SiO_2_ and Si/Al_2_O_3_ chips respectively. As an example, Figure 7 shows the top surface of the modified microcantilever with MCM-48 spherical particles. A large top surface coverage of MCM-48 is attained with tightly packed arrangements preferentially observed on the passivated Al_2_O_3_ coil. By contrast, a remarkably enhanced surface coverage is obtained for CBV100 crystals when using a similar approach (see Figure 8a–d). This observation agrees with the comparatively higher hydrophilic character of CVB100, i.e. higher concentration of hydroxyl groups on the crystal surface due to its low Si/Al ratio. Thus, the interactions with the plasma-activated surface of the chip are notably promoted. However, the majority of these chips were failed in the pre-conditioning stage carried out in the measuring set-up due to thermal coefficient mismatch and abrupt water degassing. A similar behavior is observed, although less exacerbated when using electrostatic assisted deposition approach by poly-(diallyldimethylammonium chloride) (PDDA) as the intermediate cationic polyelectrolyte (see Figure 8e–h). 

#### 2.2.3. Functionalization Method 2: Direct Spotting 

The sciFLEXARRAYER S1 micro-array spotter, from Scienion AG (Berlin, Germany), was used for the coating of cantilever probes. Such an instrument is a non–contact piezo-dispensing system, which allows spotting and dispense of liquids in the pico- to nanoliter range. This methodology offers many advantages: automatic and rapid spotting process, reproducible coatings for a given dispensing solution and target, no damage to sensitive surfaces and easily scalable to high-throughput production. The probe solutions were the following: Ethanolic suspension 2% wt. of MCM-48 nanoparticles.Aqueous suspension 1% wt. of CBV100 crystals. A previous separation by centrifugation is carried out to reduce the particle size below 700 nm.Aqueous suspension of PDDA 0.2% wt. was eventually used as intermediate cationic polyelectrolyte to improve the coverage of the surface with MCM-48 spherical nanoparticles by electrostatic interactions.

Table 2 gives an overview of all microcantilevers prepared for this study. A total number of 6 cantilevers have been functionalized with both hydrophilic materials, with mass loadings varying from 15 to 73 ng (calculated from the recorded Δf assuming rigid solid behavior of the by-layer beam). The optical images of the spotting coated cantilevers are shown in Figure 9. As with the previous functionalization method, the coating with CBV-100 material on #162 chip (see Figure 9d) appears denser than MCM-48 counterparts even using less concentrated suspensions. Once again, this effect is attributed to the higher hydrophilic character of CVB100 and superior affinity towards the passivated top surface of the Si beam. On #188 and #144 chips, both coated with MCM48 (see Figure 9a and Figure 9b, respectively), the sensing material is preferentially allocated on the lever edges. Thus, in an attempt to improve the wetting of the microcantilever top surface, PDDA-assisted coating was tested out on an MCM-48 coated chip #163. As can be observed in Figure 9c, this intermediate layer notably improves the homogeneity of the coating, leading to a uniform distribution of MCM48 over the whole tip. Herein, it is worth pointing out that PDDA layers could ultimately modify the mechanical behavior of the functionalized beam due to the well-known plastic and viscous properties of polymer films. 

## 3. Results

### 3.1. Density and Viscosity Measurement Results

For a standard density and viscosity calibration of a bare cantilever 4 different gases (N_2_, CO_2_, Ar and He) are measured at temperatures between 0 and 60 °C and pressures between 1 and 10 bar. The sensor output is calibrated by fitting the sensor data according to the model described in Section A.1. The measurement performance achieved for six different cantilevers is listed in Table 3. In addition, the measuring data of one cantilever is shown as an example in Figure 10. The density and viscosity deviations (at a 95% confidence level) are in the range of 0.023 to 0.056 kg/m^3^ and 0.18 to 0.38 µPa·s, respectively; what corresponds to about 1 % to 2% of reading value for the investigated gases (see Figure 10).

Most of the typical industrial gases can be reliably distinguished from density and viscosity, temperatures at a given pressure and temperature with the help of models or properties data bank. In this work, National Institute of Standards and Technology Reference Fluid Thermodynamic and Transport Properties (NIST Refprop) Database [19] is used for such purposes. This is illustrated in Figure 11 where density and viscosity data of different binary mixtures at standard conditions (i.e., 20 °C and 1.013 bar) calculated from experimental values of $f_{0}$ and Q according to Section A.1, are plotted against each other. Yellow diamonds mark theoretical density and viscosity values of various pure gases taken from [19]. The grey dotted lines mark the course of theoretical density and viscosity when mixing the single components. Red dots mark hypothetical measurement examples for different binary mixtures—the extension of the dots corresponds approximately to the expected measurement uncertainty. Thanks to viscosity estimation provided by the model, this approach also works for mixtures with gases of similar molecular weight, such as argon and CO_2_. Without this additional information all mixtures of Ar and CO_2_ could be badly distinguished and the Ar-70%/He-30%, Ar-72%/H_2_-28% and N_2_-90%/CO_2_-10% mixtures could hardly be distinguished.

### 3.2. Humidity Measurement Results

The experiments with humidity at ppm level were carried out either with standard gas mixtures (argon with 100 ppm water content) or with mixtures of dry and ambient air which was flushed through the measuring chamber at different flow rates. By adjusting the volume flow rate of the dry air from 30 scc/min to 100 scc/min the humidity in the measuring chamber could be varied. A reference dew point sensor in the measuring chamber always monitored the humidity (dew point) with a measurement uncertainty of 0.4 °C [15]. From the dew point and the gas temperature, H_2_O partial pressure or vapor concentration can be calculated to confirm the water dilution ratio. 

#### 3.2.1. Typical Response of Cantilevers in the Presence of Humidity

Figure 12 shows examples of typical measurements with the multiparameter gas system. After several heating cycles up to temperatures >200 °C, the sensing material can be considered as fully degassed. When exposed to a humid gas, in this case argon with vapor concentration of 100 ppmV, the frequency of the coated cantilevers drifts slowly downwards due to the adsorption of water molecules and the associated increase in mass according to Equation (2). By contrast, the frequency drift of the pristine cantilever in the same experiment is practically negligible. As it can be seen, the adsorption process is rather slow. It can take several hours until equilibrium is reached. This observation is attributed to the fluido-dynamic conditions, i.e., laminar regime Re = 1. The experimental gas velocity in the measuring chamber is circa 1 mm/s, i.e. three orders of magnitude lower than the typical values commonly used [10,20,21] with response time values lower than 3 min. That is why further efforts are being devoted on the gas measuring chamber for a practical implementation.

The frequency shift of the cantilever when exposed to argon with 100 ppmV H_2_O at 29 °C is smaller for the MCM-48 coated #166 chip (−30 Hz after 270 min) than for the CBV100-coated #181 chip at the same temperature (−50 Hz after 90 min). This observation is supported by the different sensing material loadings, i.e. 15 ng vs 48 ng for #166 and #181 chips, respectively. Unexpectedly, the influence of the working temperature on the resonance frequency is almost negligible (see Figure 12). Apparently, this observation contradicts the usual Langmuir adsorption theory [22,23,24] and the findings from the isotherm experiments (Figure 6). In general, the sorption amount at equilibrium conditions decreases with temperature due to the exothermicity of the adsorption process. On the other hand, the apparent sorption kinetics on thicker sorbent films improves with temperature due to the diffusion of the adsorbed species from the external surface to the internal nanoporous network is the controlling step. However, this seems not to be the case in our sorptive surfaces due to their thickness values are commonly below 1 micron (see Figure 8g,h). Our explanation relies on a combined mass-stress effect due to water adsorption. Since we are using very thin cantilevers in the range of 2.5 to 4.5 µm and the coating layer can also have a thickness of 0.1 to 1 µm or more, the internal stiffness of the coating layer has to be considered as well. This means the frequency drift cannot be explained by a mass effect only, but it will most probably also be a stiffness effect involved [25]. Baimpos et al. [26] describes, for example, a change in the elastic properties i.e., the Young’s modulus of zeolite films upon molecular adsorption. This effect was also determined to be temperature-dependent. In our case such an effect could amplify, weaken or even dominate the mass influence on frequency. See for details of the theory Section A.3 and also the discussion of sensing approach 2.

#### 3.2.2. Sensing Approach 1: Dynamic Response after Degassing

We propose monitoring the dynamic behavior of a degassed microcantilever, i.e., immediately after the heating cycle, when exposed to a humid gas. The working principle is to measure the initial frequency change rate, i.e. up to 3–10 min, and to correlate this rate with the vapor concentration. According to the gas adsorption model on nanoporous solids explained in Section A.2., the evolution of the effective mass of the beam due to water adsorption $\Delta m_{H2O}$ is described as follows:(1)$$\Delta m_{H2O}\left(t\right)=m_{H2O}{}_{eq}\left(1-e^{-k_{ad}^{\prime }\left(t-t_{0}\right)}\right),$$
and these mass changes are correlated with the recorded resonance frequency shifts:(2)$$\Delta f\left(t\right)=f_{0}\left[-\frac{1}{2m_{tot}}\right]\Delta m_{H2O}\left(t\right)=f_{0}\left[-\frac{m_{H2O}{}_{eq}\left(1-e^{-k_{ad}^{\prime }\left(t-t_{0}\right)}\right)}{2m_{tot}}\right],$$
with the net sorption rate constant, $k_{ad}^{\prime }$, the total cantilever mass, $m_{tot}$, and the time, $t_{0}$, where the cantilever is completely degassed and the mass of the adsorbed water molecules, $m_{H2O}{}_{t_{0}}$, is zero.

By adopting these equations, the nanoparticles are considered as single non-adherent mass points on the cantilever surface. Thus, the Young’s modulus of the functionalized microcantilever and the uncoated reference are supposed identical. Furthermore, due to the exothermicity of the adsorption process (see Figure 6 and Appendix B), the amount of adsorbed water molecules on the nanoporous materials would decrease with increasing temperature. 

Following this approach, the monitoring routine would imply a periodic repetition of this sequence: degassing by heating (desorption), measuring (adsorption), degassing by heating (desorption). An illustrative example is shown in Figure 13 for MCM-48 coated microcantilever. Thus, an initial frequency change rate of −0.3 Hz/min is experimentally evaluated immediately after the 4^th^ heating cycle when exposed to 190 ppmV of water in air. To test the validity of the model, we fitted the #166 chip experimental data, shown in Figure 13, to Equation (2) but using two different boundary conditions denoted as model fit 1 and model fit 2, respectively. In model fit 1 a complete degassing of MCM48 is assumed at $t_{0}$ = 93 min; and in model fit 2, the same conditions are assumed at $t_{0}$ = 0. The net adsorption constant, $k_{ad}^{\prime }$, calculated from both fittings is rather similar, 0.00028 s^−1^. By contrast, significant variations in the amount of adsorbed water at equilibrium, $m_{H2O}{}_{eq}/m_{NP},$ are encountered: 5 mg/g and 22 mg/g for model fit 1 and model fit 2, respectively. We attribute these discrepancies to an incomplete recovery of the sorption capacity of the MCM-48 due to the degassing procedure is not properly accomplished with our prototype. Accordingly, the registered frequency change rate and the total frequency step could be significantly less than theoretically expected due to the adsorption driven force and adsorption capacity at working conditions are diminished (see also Equation (A10)). These limitations are somehow hindering the applicability of this sensing approach. 

#### 3.2.3. Sensing Approach 2: Instantaneous Response without Previous Degassing

Unlike the previous Sensing Approach 1, the functionalized microcantilever is herein assumed to be a bilayer cantilever due to the thickness of the nanoporous sensing layer, and the mass and stiffness of the nanoporous sensing layer is not negligible compared to the Si beam. Thus, the microcantilever consists of a silicon support layer and a covering layer of nanoporous particles submicrometric in size (see Figure A1). In addition, the Young’s modulus of the nanoporous material is considered dependant on both, the temperature and the occupancy of the adsorption sites. Thus, the frequency change due to an increase of adsorbed amount of water molecules can be rewritten as follows (see Section A.2 and A.3 for model details):(3)$$\Delta f\left(t\right)=f_{0}\left[-\frac{1}{2m_{tot}}+\frac{1}{2E_{tot}}\frac{\alpha_{E}}{m_{NP}}\frac{h_{NP}}{\left(h_{NP}+h_{Si}\right)}\right]\Delta m_{H2O}\left(t\right).$$

According to Equation (3), there is a numerical solution where the frequency change becomes zero or even positive with increasing effective mass. The latter is exactly what we encountered with cantilevers functionalized with CBV100 crystals (see Figure 8g and Figure 9d). Because of the thickness of the zeolite coatings, always superior than MCM-48 counterparts, and the higher water sorption heat (see Appendix B), the release of the zeolite nanopores could eventually be incomplete after degassing conditions. An alternative working mode to circumvent such constrains is the so-called “instantaneous response without previous degassing”. 

Following this sensing approach, the resonating microcantilevers are just left equilibrating with the environment without intermediate heating steps. Surprisingly, the measurements showed a direct correlation of frequency with the vapor concentration of the environment (Figure 14). This experimental observation is supporting a dominant stiffness effect, since the mass effect would play in the opposite direction. 

As can be observed in Figure 14, the proposed Equation (3) (see Section A.3. for more details) is able to predict the frequency response of the CBV100-coated #162 chip. The model parameters compiled in Table 4 are those obtained by least square fitting of such experimental data. However, it should be noted that the set of parameters determined by the model depends on the sorbent loading. This initial coefficient notably influences the subsequent solutions. The goodness of the fitting is identical for Set 1 to Set 5 (1σ = 104 Hz). In all the cases, model fittings leading to physically reasonable parameters are found in a range between 15 ng to 75 ng of CBV100 loading. Thus, the Young’s modulus of CBV100 increases with the adsorbed amount of water molecules ($\alpha_{E}$ > 0). As consequence, the sensitivity of the sensor for water detection, herein defined as frequency change per concentration unit (Hz/ppmV), decreases with increasing temperature as experimentally observed. It is noteworthy that additional efforts are still required, more cantilevers and a wider range of experimental conditions have to be measured, in order to obtain the set of model parameters for CBV100 and MCM-48 sensing material that correlates the mechanical response of coated cantilevers with the water content for the on-field application.

Assuming that the mass effect is damped by a superimposed adsorption-dependent stiffness effect according to Equation (3), we can now try to explain the unexpected temperature behaviour shown in Figure 12 for the MCM-48 coated #166 chip. It is not surprising that the two effects behave differently over temperature. At room temperature, we still have a dominant mass effect, but a lower sensitivity than expected from the mass effect alone. The relative influence of stiffness shifts over temperature, so that it becomes lower at higher temperatures and higher at lower temperatures. This would lead to a temperature at which the two effects compensate each other and no sensitivity at all can be observed. This effect occurs at about 10 to 15 °C for the CBV100-coated #162 chip.

To extend the practical application of the gas-monitoring system, the influence of fluid density or process pressure on the functionalized cantilever frequency has been incorporated to the model. As already shown in Section 3.1, Table 3, the sensitivity ∆f/∆ρ of a bare cantilever is −190 Hz/(kg/m^3^). Considering that coated cantilever would behave in a similar way than the uncoated analogue, Equation (3) could be extended as follows: (4)$$\Delta f\left(t\right)=f_{0}\left[-\frac{1}{2m_{tot}}+\frac{1}{2E_{tot}}\frac{\alpha_{E}}{m_{NP}}\frac{h_{NP}}{\left(h_{NP}+h_{Si}\right)}\right]\Delta m_{H2O}\left(t\right)+\frac{df}{d\rho_{fl}}\Delta \rho_{fl}\left(t\right).$$

The final step would be to build an inverse model which provides the water partial pressure or vapor concentration from the frequency values of functionalized and non-functionalized microcantilevers and temperature and pressure sensor outputs. 

## 4. Conclusions

In this study, we have shown the capabilities of the multiparameter gas-monitoring system prototype to measure the gas density and viscosity with relative accuracy in the range of 1% to 2% under real pressure and temperature conditions. Similarly, the concentration of the individual components in binary as well as ternary gas mixtures is determined with the help of the NIST Refprop Database [19]. This means that most of the common process gas mixtures, e.g., welding shielding gases or packaging gases in the food industry, can be clearly monitored.

In a further step, the combination of non-functionalized and functionalized resonating microcantilevers in the single sensing platform has enabled the humidity monitoring at ppm level under non-ideal conditions, i.e., with fluctuations on temperature and pressure. The functionalization protocole with hydrophilic nanoporous materials, i.e., microporous Y type zeolite and mesoporous MCM-48 silica particles, has been defined following reproducibility, homogeneity, procurement and scale-up criteria. All the coatings on the SCL-Sensor.Tech tips have been performed with a non–contact piezo-dispensing system. The optimized functionalization process involves the deployment of PDDA as cationic polyelectrolyte on the top surface to promote homogeneous coverage of the beam. 

The mechanical response of functionalized cantilevers when exposed to humidity at ppm level reveals that the water adsorption on the nanoporous coating causes not only a change in effective mass of the beam but also induces tensile-compressive stress. Among the different sensing strategies proposed for humidity monitoring, the so called “instantaneous response without previous degassing” seems more adequate for practical implementation. Following this working mode, there is a direct correlation of the resonance frequency with the water vapor concentration in the ppmV range.

Above all, these preliminary results are paving the way for further improvements of the prototype (measuring chamber miniaturization, heating unit robustness, roughness of the SCL-Sensor.Tech top surface) and provide guidelines for additional tests and calibration routines. The knowledge gained in this study supports the feasibility of our multiparameter gas-monitoring platform for synthetic welding mixtures under real process conditions.

## Figures and Tables

**Figure 1 micromachines-11-00283-f001:**
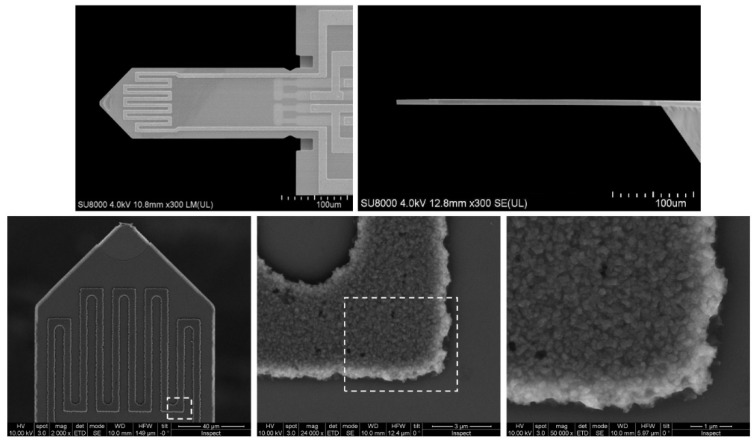
**Top**: top and side view of the silicon microcantilevers from SCL-Sensor.Tech. (PRSA-L300-F50-TL-PCB) [7] used in this study. The cantilevers have a length of 300 µm a width of 110 µm and a thickness between 2.5 to 4 µm. **Bottom**: scanning electron microscope (SEM) images of the cantilever top surface and details of the heater coil.

**Figure 2 micromachines-11-00283-f002:**
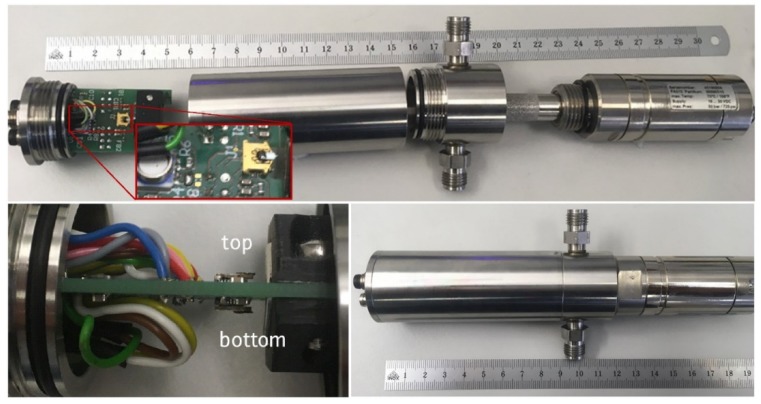
**Top**: General view of the measuring chamber with sensor printed circuit board (PCB), gas cylindrical chamber with inner diameter of 30 mm, fluidic connections and dew point sensor. **Zoom**: top view of the sensor PCB with the pressure and temperature sensor [11] and a first microcantilever in front of a permanent magnet. **Bottom left**: the second cantilever is placed on the backside of the PCB. Actuation occurs by supplying a small alternating current (AC) intensity over the metal coil on the cantilever tip. **Bottom right**: sensor PCB and dew point sensor mounted in a pressure tight measuring chamber (numbers in the scale correspond to cm).

**Figure 3 micromachines-11-00283-f003:**
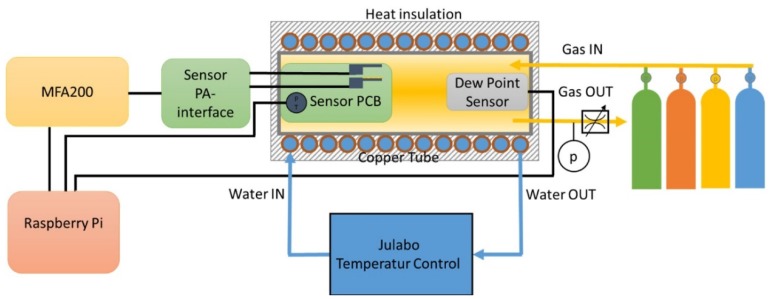
Schematic representation of the measurement setup.

**Figure 4 micromachines-11-00283-f004:**
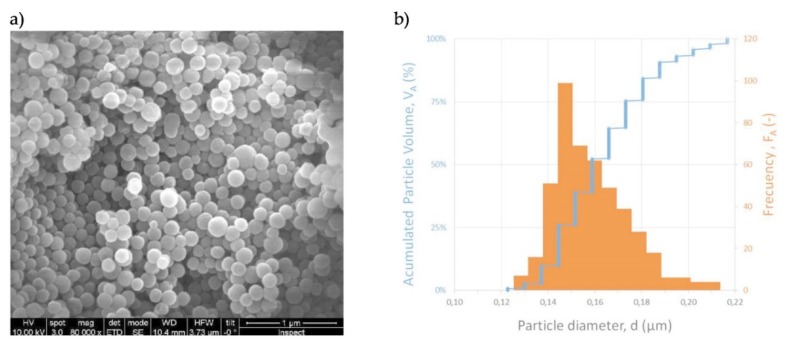
(**a**) SEM images and (**b**) digital image processing analysis for size distribution of synthetic MCM-48 spherical particles used for the functionalization of the cantilevers.

**Figure 5 micromachines-11-00283-f005:**
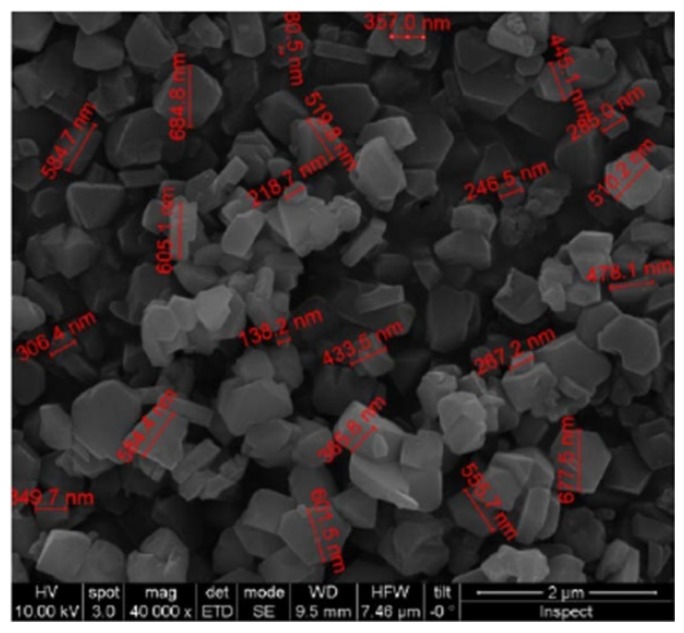
SEM images of the CBV100 crystals used for the functionalization of the cantilevers.

**Figure 6 micromachines-11-00283-f006:**
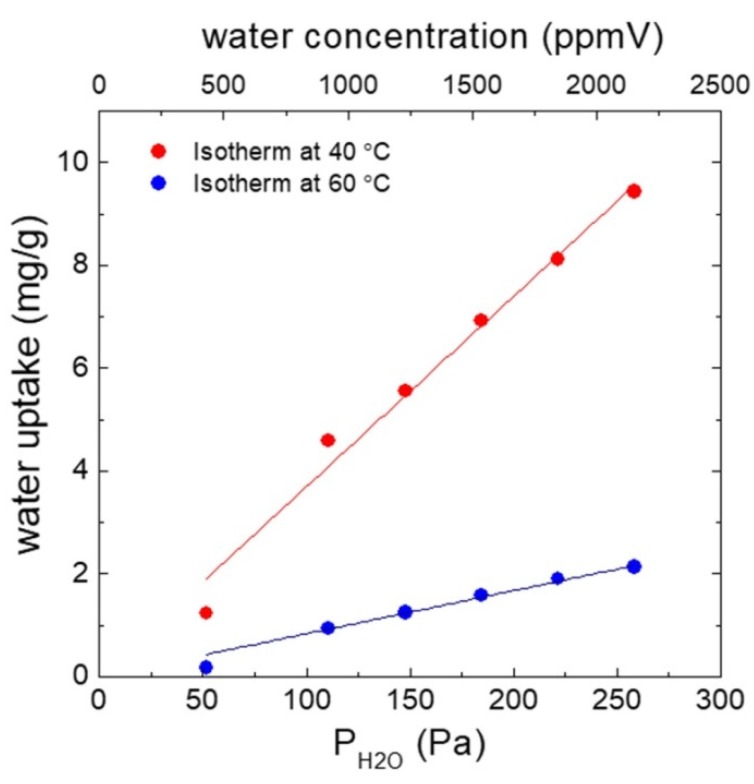
Water sorption isotherms at 40 °C and 60 °C for MCM-48 type material.

**Figure 7 micromachines-11-00283-f007:**
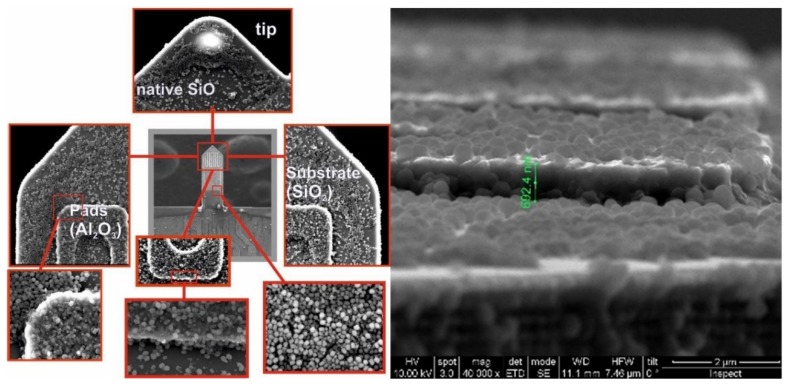
SEM images of the mesoporous silica coating on the top of the microcantilever obtained by self-assembly of MCM-48 nanoparticles onto plasma activated surfaces (cantilever #166).

**Figure 8 micromachines-11-00283-f008:**
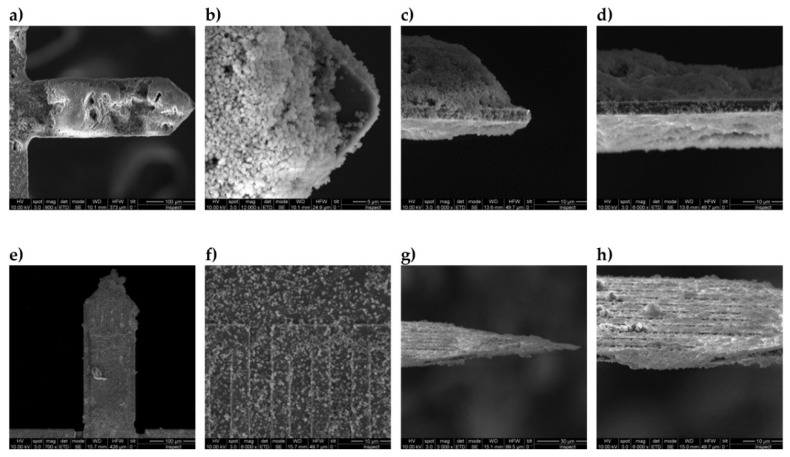
SEM images of microporous silicalite coating on the top of the microcantilever obtained by self-assembly of CBV100 crystals (2% wt. aqueous suspension) onto plasma activated surfaces (**a**–**d**) and onto poly-(diallyldimethylammonium chloride) (PDDA) activated surfaces (**e**–**h**) (cantilever #181).

**Figure 9 micromachines-11-00283-f009:**
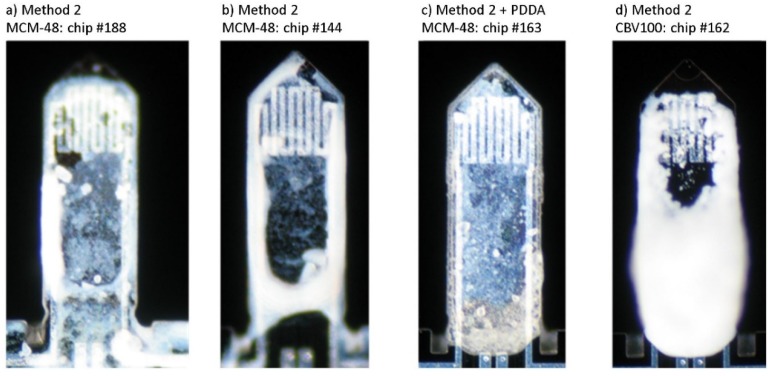
Optical microscope images of cantilevers coated by spotting of a suspension of nanoparticles directly onto the cantilever top surface according to method 2. (**a**) MCM-48 on chip #188. (**b**) MCM-48 on chip #144. (**c**) PDDA and MCM-48 on chip #163. (**d**) CBV100 on chip #162.

**Figure 10 micromachines-11-00283-f010:**
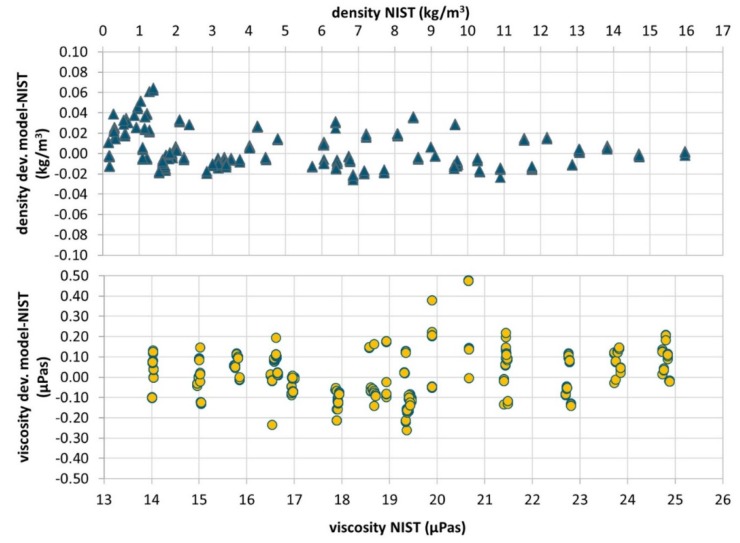
Density and viscosity measuring data of cantilever #149 calibrated according to the model described in Section A.1. Measurements for 4 gases (N_2_, CO_2_, Ar and He) at temperatures between 4.5 and 60 °C and pressures from 1 to 10 bar are shown. Deviations of the model estimations from the theoretical values from NIST Refprop Database [19] are plotted. Viscosity data not includes the measurements with helium at pressures lower than 6 bar, where the proposed model for viscosity does not work.

**Figure 11 micromachines-11-00283-f011:**
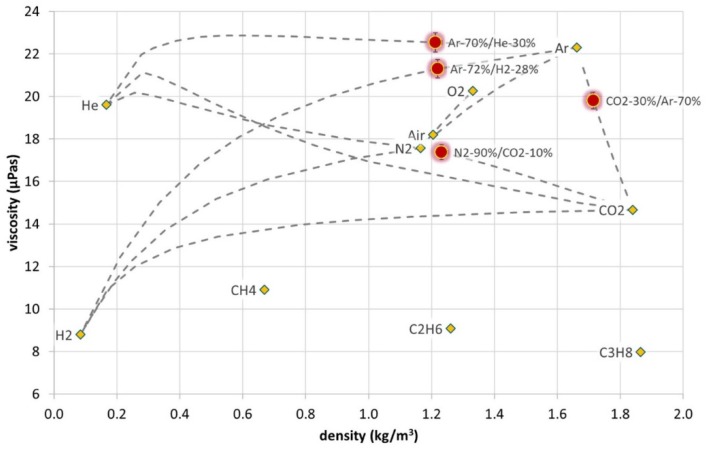
Estimation of the binary gas composition from density and viscosity values given by the sensor model described in Section A.1. and the NIST Refprop Database [19].

**Figure 12 micromachines-11-00283-f012:**
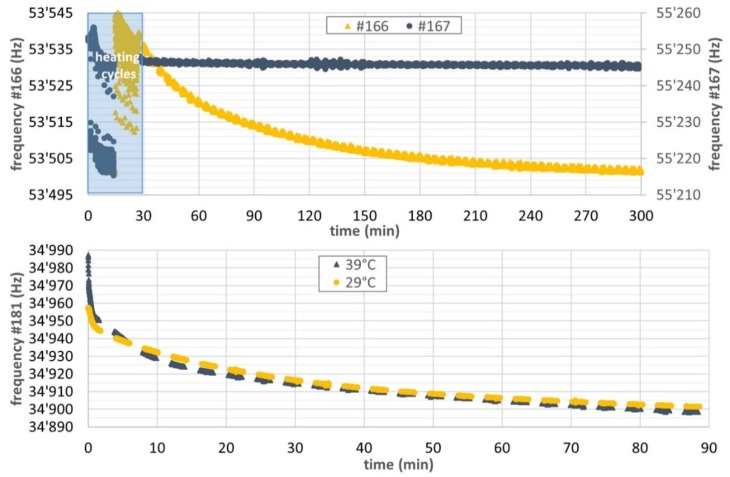
**Top**: Frequency responses of the MCM-48 coated cantilever #166 and a pristine cantilever # 167 exposed to 30 scc/min of Ar with 100 ppmV H_2_O at 29 °C and 990 mbar. **Bottom**: CBV100 coated Cantilever # 181 exposed to argon with 100 ppmV H_2_O at 990 mbar, 29 °C and 39 °C.

**Figure 13 micromachines-11-00283-f013:**
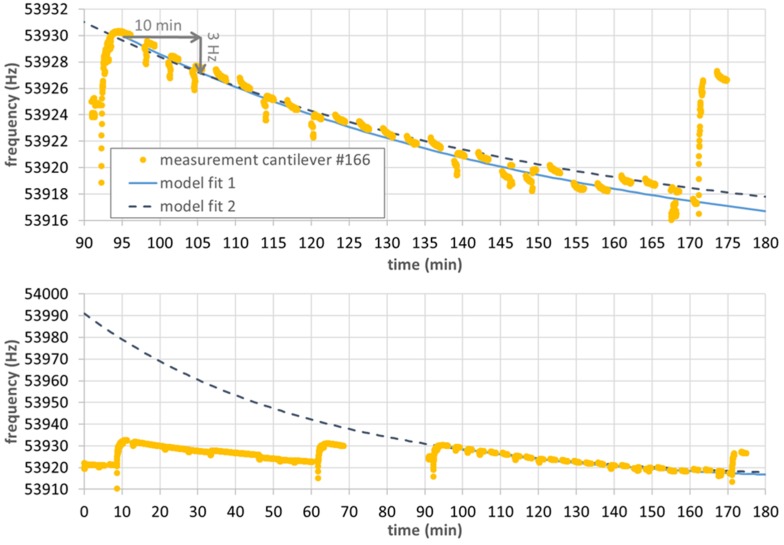
Frequency response of #166 functionalized microcantilever at 27 °C and 970 mbar exposed to 100 scc/min of air with a vapor concentration of 190 ppmV with two different model fittings both according to approach 1 Equation (2). **Top**: model fit 1 assuming a complete degassing of MCM-48 at $t_{0}$ = 93 min. **Bottom**: model fit 2 assuming a complete degassing of MCM-48 a $t_{0}$ = 0 min.

**Figure 14 micromachines-11-00283-f014:**
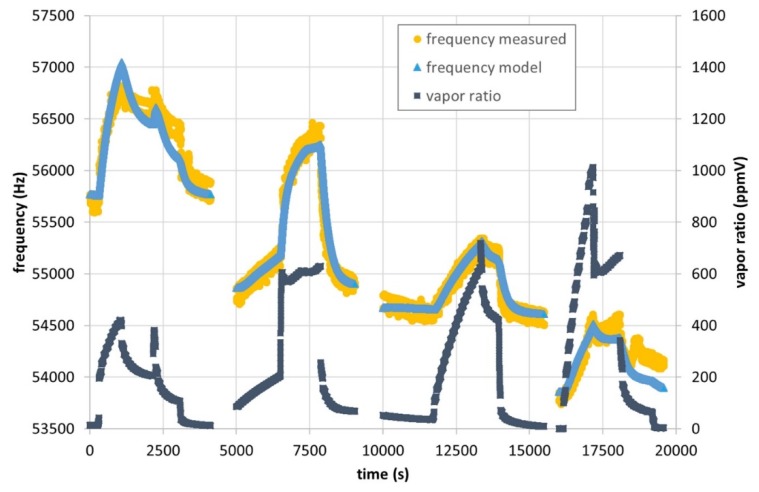
Frequency response of #162 functionalized microcantilever exposed to humid air (up to 1000 ppmV of water) and model fit with the parameters shown in Table 4 Set 2. Temperatures 16 °C (0 to 4100 s), 25 °C (5000 to 9000 s), 45 °C (10,000 to 15,500 s) and 55 °C (16,000 to 19,600 s). Pressure values between 970 and 1100 mbar.

**Table 1 micromachines-11-00283-t001:** Main properties of the commercial NaY type zeolite (CBV100) and mesoporous silica (MCM-48) used in this work.

Material	BET Surface Area	Pore Volume	Pore Size	Particle Size
	m^2^/g	cm^3^/g	Å	nm
CBV100	758	0.247	6.9	<700
MCM-48	1233	0.87	32	164 ± 18

**Table 2 micromachines-11-00283-t002:** Main characteristics of the microcantilevers measured in this study.

Chip#	Coating Material	Coating Method	Frequency in air before/after Coating (kHz)		Thickness of Cantilever Tip (µm)	Mass of Sorbent * (ng)
**29**		bare chip	35.02		2.6	0
125		bare chip	55.85		4.2	0
126		bare chip	59.93		4.5	0
149	-	bare chip	55.05		4.1	0
167	-	bare chip	55.25		4.1	0
190		bare chip	31.62		2.4	0
204		bare chip	30.45		2.3	0
166	MCM-48	Method 1 aqueous 2% wt.	54.81	53.49	4.1	15
144	MCM-48	Method 2 ethanolic 2% wt.	56.50	50.95	4.3	73
188	MCM-48	Method 2 ethanolic 2% wt.	39.37	35.02	3.0	58
163	MCM-48 on PDDA	Method 2 ethanolic 2% wt. with PDDA	56.94	52.1	4.3	62
181	CBV100	Method 1 with PDDA	38.03	34.37	2.9	48
162	CBV100	Method 2 aqueous 1% wt.	55.22	53.30	4.1	23

* Loading of sorbent material estimated from recorded frequency shift after coating according to Equation (A25). The variation of the stiffness of the cantilever due to sensitive layer deposition was not considered.

**Table 3 micromachines-11-00283-t003:** Density and viscosity measuring performance of bare cantilevers after calibration with N_2_, CO_2_, Ar and He at temperatures between 0 and 60 °C and pressures between 1 and 10 bar.

Chip#	Sensitivity ∆f/∆ρ	Density Deviation ^1^	Viscosity Deviation ^1,2^ relative absolute		Q_factor__min	Q_factor__max
	Hz /(kg/m^3^)	kg/m^3^	%	µPa·s		
29	−191	0.038	2.0	0.38	40	230
125	−189	0.023	1.4	0.25	90	500
126	−195	0.035	1.0	0.18	90	550
149	−192	0.028	1.1	0.20	70	490
190	−191	0.039	1.5	0.28	30	200
204	−183	0.056	1.4	0.27	30	200
mean	−190	0.037	1.4	0.26	58	362
min	−183	0.056	2.0	0.38	90	550
max	−195	0.023	1.0	0.18	30	200

^1^ confidence level of 95% (2 sigma) ^2^ when density >0.9 kg/m^3^ (i.e. not including helium at pressures lower than 6 bar where the proposed model for viscosity does not work).

**Table 4 micromachines-11-00283-t004:** Different sets of coefficients for the model fit of CBV100-coated #162 chip (coefficients $m_{Si}$ to $T_{0}$ are kept constant).

Coefficient	Set 1	Set 2	Set 3	Set 4	Set 5	Unit
1σ	104	104	104	104	104	Hz
$m_{Si}$	311	311	311	311	311	ng
$E_{Si}$	170	170	170	170	170	GPa
$\rho_{Si}$	2330	2330	2330	2330	2330	kg/m^3^
$h_{Si}$	4.16	4.16	4.16	4.16	4.16	µm
$\rho_{\mathrm{NP}}$	2020	2020	2020	2020	2020	kg/m^3^
$m_{NP}$	15	50	75	50	47	ng
$h_{NP}$	0.25	0.82	1.23	0.82	0.77	µm
$T_{0}$	25	25	25	25	25	°C
Parameters calculated by the model						
$k_{ad}^{\prime }{}_{0}$	4.89 × 10^−3^	4.89 × 10^−3^	4.89 × 10^−3^	4.86 × 10^−3^	4.85 × 10^−3^	1/s
$E_{NP}{}_{0}$	61	74	67	102	161	GPa
$\beta_{E}$	2.92 × 10^−^^1^	7.75 × 10^−2^	5.86 × 10^−2^	4.78 × 10^−2^	3.37 × 10^−2^	1/K
$\alpha_{E}{}_{0}$	4.18	5.89	6.35	5.31	5.21	GPa/(mg/g)
$\beta_{\alpha }$	−1.39 × 10^−2^	−1.41 × 10^−2^	−1.42 × 10^−2^	3.42 × 10^−2^	3.40 × 10^−2^	1/K
$K_{0}^{\prime }$	6.72 × 10^−^^1^	1.50 × 10^−1^	9.52 × 10^−2^	1.79× 10^−1^	2.03 × 10^−1^	mg/g·Pa^−1^
$\Delta G_{sorption}$	14.82	14.82	14.83	51.29	51.20	kJ/mol
$df/d\rho_{fl}$	−210	−210	−210	−217	−217	Hz/(kg/m^3^)

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
