# Peer review of "A Multiparameter Gas-Monitoring System Combining Functionalized and Non-Functionalized Microcantilevers"

_micromachines, 2020, doi:10.3390/mi11030283_

Round 1
Reviewer 1 Report
This study aims to develop a compact, robust and maintenance free gas concentration and humidity monitoring system for industrial use in the field of inert process gases. The authors claim a novel multiparameter gas sensor concept combining functionalized and non-functionalized resonating microcantilevers allowing the simultaneous measurement of the fluid properties density, viscosity as well as humidity under varying process conditions is presented.
However, several problems should be answered for further consideration of publication in Micromachines.
Can the authors explain the novelty of this work compared with the following reference?
Abedinov, N., et al. "Chemical recognition based on micromachined silicon cantilever array." Journal of Vacuum Science & Technology B: Microelectronics and Nanometer Structures Processing, Measurement, and Phenomena 21.6 (2003): 2931-2936. There are no scale bars in images of Figure 2. In the manuscript, the reproducibility of the sensor did not be demonstrated nither for density, viscosity as well as humidity. The explanation of Figure 11 is not clear, it is very difficult to understand. The entire article is full of data, but there is no proper description and discussion, making it very unreadable for readers.
Reviewer 2 Report
The main problem I can see in the approach 1 is that they took hours time to see the equilibrium. I guess why the PCB or some element in the chamber suck the water to be equilibrium condition in the chamber.
I would recommend to remove this first approach from the manuscript.
The 2nd approach is interesting to see quick response as dynamic response.
However the model explanation is unclear.
Reviewer 3 Report
This manuscript reports a coated microcantilever sensor system for monitoring gas composition for inert gas mixtures. The authors have clearly explained the physical construction of the devices and the basic operating principles of the system when discriminating different gases in simple mixtures. The authors have provided sufficient empirical data to support the use of the sensor to measure changes in gas density and viscosity and so therefore have a basis for assessing changes in composition of simple mixtures of known gases. However, the authors then go on to assert an ability to measure humidity down to 100 ppmV based on supposed changes in cantilever frequency with change in humidity. I do not feel that the authors have presented sufficient data to clearly support any humidity measurement ability. The data presented in Figures 14-16 has been highly over-interpreted and does not suggest a clear and explicable relationship between changing humidity and sensor frequency. The correlations in this part of the paper are of poor quality and the analysis laid out in Table 2 is highly dependent on the initial conditions used in the model. As such, I feel that a number of the final conclusions reached are unsupported and that the paper should be completely rewritten. More specific points for the authors to address are given below.
Page 1 line 34 “. . . in the % range.” This is vaguely worded. Do you mean about 1%, 10% or some value higher or lower? Be more specific with this description. Page 1 line 35 “Typical threshold . . . and 40 ppm [1].” You should specify that these thresholds pertain to welding gases. Different thresholds could be expected for other applications. Page 3 line 75 “To avoid self-promotion . . .” It is not clear exactly what you mean here. Clarify your use of the term “self-promotion”. Page 3 line 99 “. . . and reset the humidity measurement.” There is significant doubt about whether your heating step is consistently resetting the humidity conditions inside the sensor. As such, this does cast doubt on whether you can reliably acquire humidity-related sensor data using this approach. Table 1. From where were the BET surface area, pore volume and pore size parameters acquired? If they are literature values then provide a reference. If they are empirically determined then make this clear in the manuscript. Figure 9. There is still a worrying degree of variability in the thickness and extent of the coatings on these cantilevers and this would appear to contribute to variability in your later sensor output results. This is a significant disadvantage that deserves some more discussion. Page 8 lines 185-190 “In general, the . . . gas sensing applications.” From which data are you drawing these conclusions? The data in Table 2 do not address homogeneity and do not provide clear guidance on density or thickness trends. These conclusions should be reconsidered. Page 9 lines 211-214 “Furthermore, even more . . . as described in [8].” You haven’t provided any clear data suggesting that this correlation approach could be successfully used. Applying such a correlation would be expected to introduce significant error into the results that would need to be understood and quantified. Figure 12. In the figure legend and main text you mention 28 oC but in the label of the figure you give 29 o Clarify which temperature is used. Figure 13 legend. You should specify that it is a water vapor concentration used here. Figure 14. There are not clear correlations in this data either for sensitivity versus vapor ratio or versus temperature. The only scatter that comes close to a linear correlation is that for the 25 ppmV case in the sensitivity versus temperature plot but even then there are not many data points. It is therefore not valid to include linear correlation lines, particularly given you have not quoted the associated R2 Page 13 lines 282-309 “Two conclusions can . . . sensing approach 2.” Further to comment 11, these conclusions are unsound as they are based on an over-interpretation of Figure 14, inferring relationships between parameters that haven’t been proven to exist. I do not feel that the sensors are reliably detecting changes in humidity based on this data or that you can draw conclusions relating sensitivity to vapour concentration or temperature. Page 13 lines 300-301 and Figure 12. “All our cantilevers . . . Figure 12).” The differences in the plots for the two temperatures in Figure 12 are quite marginal and you have not demonstrated that this is a repeatable difference and thus statistically significant. Table 2. I feel there is too much uncertainty introduced to this analysis by the starting conditions and so the value of this analysis seems quite limited. Page 15 lines 343-344 “With this knowledge . . . in approach 1.” Given that you cannot draw firm conclusions from the data in Figure 14, I feel that this analysis is invalid. Figure 16. This plot has a poor R2 value and very large errors associated with it. I do not feel that you can reliably infer either a linear correlation or a slope value or this plot. Page 16 lines 376-378 “. . . which is capable . . . should be feasible.” It is not clear on what basis you are inferring a measurement resolution of < 100 ppmV. Such conclusions need to be clearly linked to the data provided. Page 17 lines 407-408 “The sensitivity . . . range 100 ppmV.” Where is the evidence for such a sensitivity? To quote this figure you need to be able to demonstrate that the sensor can reliably detect down to this concentration.Author Response
Please see the attachment

Round 2
Reviewer 1 Report
The authors answered all the questions and gave a proper revision of the manuscript. The revised manuscript explains the experimental methods well.Reviewer 2 Report
The manuscript show well explanations for their experimental approaches.
The mechanical response of microcantilevers has been well modelled considering both effects and the simulated results.
Reviewer 3 Report
The authors have made extensive changes to this manuscript in response to my earlier review comments. The authors have removed poor quality figures from the manuscript, better explained the scope and purpose of the paper and have emended the text to remove unsound conclusions and analysis. The authors have added some additional data to provide clarification on some aspects of the device performance and have emphasised points where more research is needed to better define the exact relationship between the sensor signal output and humidity. Taking these factors into consideration, I feel that the authors have made sufficient changes to the manuscript to merit its publication.